# Bayesian robust symmetric regression for medical data with heavy-tailed errors and censoring

Mehmet Ali Cengiz[ID][1]*, Talat Şenel[2], Muhammed Kara[ID][3]

**1** Department of Mathematics and Statistics, College of Science, Imam Mohammad Ibn Saud Islamic University (IMSIU), Riyadh, Saudi Arabia, **2** Department of Statistics, Faculty of Science, Ondokuz Mayıs University, Samsun, Turkey, **3** Department of Educational Sciences, Faculty of Education, Ondokuz Mayıs University, Samsun, Turkey

\* mamcengiz@imamu.edu.sa

## Abstract

Bayesian symmetric regression offers a principled framework for modeling data characterized by heavy-tailed errors and censoring, both of which are frequently encountered in medical research. Classical regression methods often yield unreliable results in the presence of outliers or incomplete observations, as commonly seen in clinical and survival data. To address these limitations, we develop a robust Bayesian regression model that incorporates symmetric error distributions such as the Student-t and Cauchy, providing improved resistance to extreme values. The model also explicitly accounts for both right and left censoring through its likelihood structure. Inference is performed using Markov Chain Monte Carlo (MCMC), allowing for accurate estimation of uncertainty. The proposed approach is validated through simulation studies and two real-world medical applications: lung cancer survival analysis and hospital stay duration modeling. Results indicate that the model consistently outperforms traditional methods when dealing with noisy, censored, and non-Gaussian data, highlighting its potential for broad use in medical statistics and health outcome research.

## Introduction

Regression models are a cornerstone of data analysis, often used to explore how a response variable relates to a set of predictors. Classical linear regression models are built upon two fundamental assumptions: that the error terms follow a normal distribution and that the dataset is complete, with no missing or censored observations. But in practice—especially in areas like medicine, economics, and environmental science—those assumptions often don't hold. In practical applications, data often exhibit two major challenges: heavy-tailed error distributions and censoring. The presence of outliers or extreme values can significantly distort parameter estimates in standard models, while failure to account for censored observations may lead to biased and unreliable inferences.

**Data availability statement:** The datasets used and analyzed during the current study are publicly available on Zenodo at the following DOI: https://doi.org/10.5281/zenodo.15735828.

**Funding:** This work was supported and funded by the Deanship of Scientific Research at Imam Mohammad Ibn Saud Islamic University (IMSIU) (grant number IMSIU-DDRSP2501).

**Competing interests:** The authors have declared that no competing interests exist.

To handle heavy-tailed errors, researchers have developed robust regression techniques. One popular approach involves using symmetric heavy-tailed distributions like the Student-t or Cauchy. These distributions help reduce the influence of outliers while keeping the model interpretable [1–3]. In the frequentist world, methods like M-estimators, L1 regression [4], and quantile regression are commonly used. Bayesian approaches take things further by incorporating prior beliefs and delivering full probabilistic inference ( [5–7].

When it comes to censored data, survival analysis and Tobit models are typical go-to methods [8–10]. These tools handle right, left, or interval censoring well— but they often assume specific distributions (like normal or exponential), which can make them fragile in the presence of outliers or heavy tails. Bayesian models have gained traction in last two decades because they offer a flexible framework for complex data, allow for prior knowledge, and deliver full posterior inference. Symmetric error distributions in Bayesian models have performed well under non-normality and contamination [11,12]. In the financial context, [13] applied Bayesian techniques with scale mixtures of normal distributions to improve robustness in heavy-tailed stochastic volatility models. Furthermore, [14] presented a Bayesian approach utilizing scale mixtures of skew-normal distributions to enhance the robustness and accuracy of censored linear regression models. [15] introduced a robust Bayesian model selection technique specifically tailored for heavy-tailed linear regression models, employing finite mixture distributions to effectively capture outliers and heterogeneity within data. Recently, [16] proposed a generalized Bayesian method for robust regression models in high-dimensional settings, accounting for serially correlated errors and predictors, which is particularly valuable in complex data structures encountered in modern statistical analysis. Still, there's a noticeable gap in the literature: not many studies tackle both heavy-tailed errors and censored data simultaneously using a Bayesian symmetric model. There's a clear demand for models that are robust to outliers, flexible in handling fat tails, and equipped to manage censoring.

Recent advances in robust Bayesian regression have introduced scalable and practically effective methods for heavy-tailed and censored data. For example, [17] Fan et al. (2022) developed a hard-thresholding approach to robust Bayesian regression resilient to adaptive adversarial corruption. In the context of censored outcomes, [18] Karlova et al. (2024) proposed a variational inference framework for Tobit-like Gaussian process models, offering scalable posterior estimation. Proven robustness properties of heavy-tailed Bayesian models were further formalized by [19] Gagnon (2025), with theoretical results for Student-t regression. Moreover, [20] Liu & Wang (2025) introduced censor-dependent variational inference (CDVI), which directly incorporates the censoring mechanism into the VI formulation. Finally, robust Bayesian model averaging with hyperbolic and Student-t error mixtures has been recently proposed by [21] De & Ghosh (2024), integrating outlier-resistance with variable selection. These contributions situate our work within a modern landscape of scalable, robust Bayesian estimation methods dealing with heavy tails, censoring, and computational efficiency.

This paper introduces a Bayesian robust symmetric regression model tailored for datasets that are both heavy-tailed and censored. It uses a flexible symmetric error distribution (such as Student-t or Cauchy) and directly incorporates right or left censoring into the likelihood. We use Markov Chain Monte Carlo (MCMC) methods for inference, which allows for full uncertainty quantification [22,23].

Comprehensive simulation studies and two representative medical applications, involving lung cancer survival analysis and hospital stay duration modeling, illustrate the effectiveness of the proposed Bayesian symmetric regression framework. The model consistently yields more accurate posterior estimates, narrower credible intervals, and superior predictive performance compared to conventional methods. Its ability to simultaneously accommodate heavy-tailed error structures and various forms of censoring distinguishes it from existing approaches in the literature. By integrating flexible symmetric error distributions with full Bayesian inference, this study provides a unified and robust solution for complex biomedical data settings where outliers, data irregularities, and partial observations are prevalent.

## Model Specification

We consider the following flexible and robust symmetric regression model:

$$y_i^* = x_i^T \beta + \varepsilon_i, \qquad i = 1, 2, \cdots, n \tag{1}$$

where $y_i^*$ is the latent (unobserved) continuous response variable, $x_i \in \mathbb{R}^p$ is the covariate vector for the $i$th observation, $\beta \in \mathbb{R}^p$ is the vector of regression coefficients and $\varepsilon_i$ is a symmetric and heavy-tailed error term.

In this study, we consider two symmetric heavy-tailed distributions for the error term:

- The Student-$t$ distribution is given as $\varepsilon_i \sim \mathcal{T}\left(0, \sigma^2, v\right)$, *with v > 2*

- The Cauchy distribution (a special case of Student-$t$ with $v = 1$) is given as $\varepsilon_i \sim Cauchy(0, \sigma)$

This modeling structure allows for flexible handling of uncertainty and variability in real-world data. In particular, it offers a principled way to address common challenges such as censoring and non-Gaussian errors in applied settings. In light of the complexities commonly observed in survival and clinical datasets such as censoring and heavy-tailed errors the following remark is introduced.

**Remark.** The proposed model flexibly handles both right and left censoring while providing robustness to heavy-tailed errors. This dual capability makes it particularly suitable for medical applications, where survival and clinical outcomes often suffer from partial observability and outliers.

This modeling framework allows us to investigate the robustness of both distributions and compare their performance under heavy-tailed data. In many practical scenarios, the outcome variable is subject to censoring. Let the observed variable $y_i$ be defined as:

$$y_i = \begin{cases} y_i^*, & \text{if } L_i < y_i^* < U_i & (\textbf{observed}) \\ L_i, & \text{if } y_i^* \leq L_i & (\textbf{left} - \textbf{censored}) \\ U_i, & \text{if } y_i^* \geq U_i & (\textbf{right} - \textbf{censored}) \end{cases} \tag{2}$$

Here, $(L_i, U_i)$ denote the observable interval for the $i$th observation. The likelihood contribution of each observation depends on its censoring status:

- For observed values: $\mathcal{L}_i = f\left(y_i \mid x_i, \beta, \sigma, v\right)$

- For left-censored values: $\mathcal{L}_i = F\left(L_i \mid x_i, \beta, \sigma, v\right)$

- For right-censored values: $\mathcal{L}_i = 1 - F\left(U_i \mid x_i, \beta, \sigma, v\right)$

where $f(\cdot)$ and $F(\cdot)$ denote the PDF and CDF of the chosen symmetric distribution (Student-$t$ or Cauchy). The full likelihood function that accounts for both observed and censored outcomes under the proposed symmetric error model is formally stated in the following proposition.

Proposition. The full likelihood function for a sample of possibly censored observations is given by $p(y | \mathbf{X}, \beta, \sigma, v) = \prod_{i=1}^{n} \mathcal{L}_i$. This formulation allows for exact handling of censoring mechanisms without relying on imputation, and seamlessly incorporates robust error modeling through the symmetric distributions.

A Bayesian framework requires prior distributions for all unknown parameters. We specify the following priors:

- Regression coefficients are $\beta \sim N(0, \tau^2 I_p)$, where $\tau^2 = 10^4$

- Scale parameter (common to both models) is $\sigma \sim \text{Half-Cauchy}(0, 5)$

- Degree of freedom for Student-$t$ is $v \sim \text{Gamma}(2, 0.1)$, with support $v > 2$

Note that for the Cauchy model, $v$ is fixed at 1. The joint posterior distribution of the parameters is given as follow:

$$p(\beta, \sigma, v | y, \mathbf{X}) \propto \prod_{i=1}^{n} \mathcal{L}_i \cdot p(\beta) \cdot p(\sigma) \cdot p(v) \tag{3}$$

where $\mathcal{L}_i$ denotes the individual likelihood contribution based on the censoring status. Note that for the Cauchy model, $v$ is not sampled and is treated as fixed.

We perform Bayesian inference using MCMC methods to draw samples from the posterior distribution. For the Student-$t$ model, a data augmentation approach is used where the $t$-distributed errors are represented as a scale mixture of normals:

$$\varepsilon_i \sim \mathcal{N}(0, \lambda_i^{-1}, \sigma^2), \qquad \lambda_i \sim \text{Gamma}\left(\frac{v}{2}, \frac{v}{2}\right) \tag{5}$$

This allows for Gibbs sampling of latent scales $\lambda_i$ and improves mixing. For the Cauchy model, a similar representation using inverse-gamma mixing is employed, or direct sampling via Hamiltonian Monte Carlo (HMC) can be used.

Posterior summaries such as posterior means, medians, and 95% credible intervals are obtained using the posterior draws. Model convergence is assessed via standard diagnostics such as trace plots, the potential scale reduction factor ($\hat{R}$), and effective sample size (ESS).

## Model Comparison

To compare the performance of the proposed Bayesian symmetric regression models under different error assumptions (Student-t vs. Cauchy), we use a range of model evaluation criteria. These methods provide a principled way to assess and compare models in terms of goodness-of-fit, predictive accuracy, and robustness to outliers and censored data.

One important diagnostic is the posterior predictive check, which evaluates how well a model replicates the observed data. This involves simulating datasets from the model's posterior predictive distribution and comparing these to the actual observed data using either graphical plots or summary statistics [24]. Significant differences between the replicated and observed data may signal model misfit.

The log predictive density (LPD) offers a direct measure of model fit by summing the log-likelihood of the observed data under the model's posterior predictive distribution. A higher LPD value suggests better in-sample predictive performance.

The Widely Applicable Information Criterion (WAIC) is a fully Bayesian alternative to AIC. WAIC adjusts for model complexity through the effective number of parameters, helping guard against overfitting. It is also asymptotically equivalent to leave-one-out cross-validation (LOO-CV) under regular conditions, making it a widely accepted tool for Bayesian model comparison [25,26]. WAIC is computed as:

$$\text{WAIC} = -2\left(l_{\text{ppd}} - p_{\text{WAIC}}\right) \tag{6}$$

where $l_{ppd}$ is the log pointwise predictive density and $p_{WAIC}$ is the estimated effective number of parameters.

LOO-CV evaluates a model's predictive performance by systematically leaving out each observation, one at a time, and calculating the log-likelihood of the omitted data given the model trained on the remaining observations. While this process can be computationally intensive, the Pareto-smoothed importance sampling (PSIS-LOO) method provides an efficient approximation using MCMC samples [27]. PSIS-LOO is especially valuable for assessing how well a model is likely to generalize to new, unseen data, making it a powerful tool for robust model validation in Bayesian frameworks.

## Simulation study

In this section, we present a simulation study to evaluate the performance of the proposed Bayesian robust symmetric regression model under various data conditions. Specifically, we investigate the behavior of the model under two different symmetric heavy-tailed error distributions—Student-t and Cauchy—and examine its robustness in the presence of outliers and right-censored data. The goal is to assess the accuracy and credibility of parameter estimates as well as the predictive performance of the model under different levels of data complexity.

The simulated data are generated from a linear regression framework with the true model defined as
$y_i^* = \beta_0 + \beta_1 x_{1i} + \beta_2 x_{2i} + \varepsilon_i$, where $x_{1i} \sim Uniform(-1, 1)$ and $x_{2i} \sim Bernoulli(0.5)$. The regression coefficients are fixed as $\beta_0 = 1$, $\beta_1 = 2$ and $\beta_2 = -1$. The error term $\varepsilon_i$ is generated either from a Student-t distribution with degrees of freedom $v \in \{3, 5, 10\}$ or from a Cauchy distribution, which corresponds to the limiting case of the Student-t with $v = 1$. The error scale is fixed at $\sigma = 1$, and the sample size is set to $n = 300$ for all scenarios.

To introduce censoring, we define an upper censoring threshold $c$ such that if $y_i^* > c$, the observed value is set to $y_i = c$. The threshold is selected to introduce approximately 20% and 40% right-censoring in the data. In the uncensored scenario, all values of $y_i$ are fully observed. This design allows us to systematically examine how censoring affects the accuracy and uncertainty of posterior estimates under different error distributions.

For each simulated dataset, we fit both the Student-t and Cauchy models using the Bayesian framework described in Section 2. Posterior inference is performed using Markov Chain Monte Carlo (MCMC) methods via the No-U-Turn Sampler (NUTS) implemented in Stan. Each model is estimated using four parallel chains, with 2,000 iterations per chain including 1,000 warm-up iterations. Convergence diagnostics such as potential scale reduction factor $\hat{R}$ and effective sample size (ESS) are monitored to ensure reliable estimation.

To evaluate model performance, we consider several metrics: the posterior mean bias of each regression coefficient, the root mean square error (RMSE), the average width of 95% credible intervals, and the coverage probability (CP), which indicates the proportion of credible intervals that contain the true parameter values. In addition, we use the widely applicable information criterion (WAIC) and Pareto-smoothed importance sampling leave-one-out cross-validation (PSIS-LOO) to compare model fit and predictive accuracy between the two error distribution specifications.

The simulation study is replicated 1000 times for each combination of error distribution and censoring level. Results are summarized across replications using average metrics and visualized through graphs and performance tables. Table 1 summarizes the posterior estimates for the Student t model under three right-censoring levels (10%, 20%, 40%).

Table 1 presents the posterior estimates for the model parameters under 10%, 20%, and 40% right-censoring rates. Increasing the censoring level leads to wider credible intervals and slightly larger estimation errors. The posterior means remain close to the true parameter values across all censoring levels, indicating robustness of the Bayesian model. Notably, the effective sample sizes (Bulk ESS and Tail ESS) are sufficiently high, and the $\hat{R}$ values are equal to 1, suggesting good convergence of the Markov chains. Among the covariates, the binary predictor x2 shows slightly more variability in its estimates, which is consistent with its limited information content. Overall, the model maintains accurate and stable inference performance despite varying degrees of censoring.

Table 2 is used to evaluate the frequentist properties of the proposed Bayesian Student-t regression model under varying right-censoring levels. Specifically, it reports simulation-based metrics such as bias, root mean square error (RMSE), coverage probability, and credible interval width. These measures provide a comprehensive assessment of the model's

**Table 1. The posterior estimates for the Student t model under varying right-censoring rates.**

| Censoring | Parameter | Estimate | Est_Error | CI_Lower | CI_Upper | $\hat{R}$ | Bulk_ESS | Tail_ESS |
|---|---|---|---|---|---|---|---|---|
| 10% | Intercept | 1,189 | 0,099 | 0,988 | 1,379 | 1 | 2591 | 1681 |
| | x1 | 2,019 | 0,14 | 1,756 | 2,299 | 1 | 2359 | 1263 |
| | x2 | −1,11 | 0,148 | −1,409 | −0,806 | 1 | 2649 | 1551 |
| 20% | Intercept | 0,933 | 0,114 | 0,709 | 1,163 | 1 | 2587 | 1094 |
| | x1 | 1,847 | 0,137 | 1,586 | 2,126 | 1 | 2120 | 1097 |
| | x2 | −0,984 | 0,151 | −1,271 | −0,685 | 1 | 2252 | 1189 |
| 40% | Intercept | 1,086 | 0,127 | 0,835 | 1,341 | 1 | 1208 | 981 |
| | x1 | 2,439 | 0,179 | 2,091 | 2,813 | 1 | 1754 | 1270 |
| | x2 | −1,019 | 0,176 | −1,362 | −0,676 | 1 | 1691 | 1457 |

**Table 2. Simulation results for the Student-t model under varying right-censoring rates.**

| Censoring | Parameter | Bias | RMSE | Coverage | CI_Width |
|---|---|---|---|---|---|
| 10% | b_Intercept | 0,009 | 0,098 | 0,98 | 0,408 |
| | b_x1 | −0,006 | 0,116 | 0,97 | 0,494 |
| | b_x2 | −0,003 | 0,15 | 0,92 | 0,572 |
| 20% | b_Intercept | −0,004 | 0,104 | 0,96 | 0,423 |
| | b_x1 | 0,033 | 0,14 | 0,94 | 0,518 |
| | b_x2 | −0,001 | 0,134 | 0,98 | 0,58 |
| 40% | b_Intercept | 0,035 | 0,136 | 0,938 | 0,525 |
| | b_x1 | 0,029 | 0,152 | 0,958 | 0,64 |
| | b_x2 | −0,035 | 0,176 | 0,938 | 0,657 |

accuracy, precision, and uncertainty quantification. The table serves to demonstrate the model's robustness and reliability across different degrees of censoring. Across all conditions for the Student-t model under three right-censoring levels (10%, 20%, 40%), the proposed Bayesian model demonstrated low bias and reasonably small RMSE values, indicating stable estimation performance.

The coverage probabilities of the 95% credible intervals were close to the nominal level, even under severe censoring. As expected, higher censoring rates led to wider credible intervals, reflecting increased uncertainty. Notably, the estimation of the binary covariate x2 exhibited slightly higher RMSE and wider intervals, which is consistent with its reduced information content. Overall, the results confirm the robustness of the model across varying degrees of censoring.

Table 3 presents the posterior estimates of the model parameters under 10%, 20%, and 40% right-censoring levels using the Cauchy error distribution. The table includes point estimates, standard errors, 95% credible intervals, and convergence diagnostics ($\hat{R}$, Bulk ESS, and Tail ESS).

The results show that even under heavy-tailed Cauchy errors, the proposed Bayesian model provides stable and accurate inference. All $\hat{R}$ values equal 1, indicating satisfactory convergence, and effective sample sizes are sufficiently large across all parameters and censoring levels. As expected, credible intervals become wider with increased censoring, yet the parameter estimates remain close to the true values, reflecting the model's robustness to both heavy-tailed errors and censoring. Table 4 reports the simulation-based performance metrics for the Bayesian regression model using Cauchy-distributed errors under varying right-censoring levels.

The results include bias, root mean square error (RMSE), 95% coverage probabilities, and credible interval widths for each parameter. As censoring increases, RMSE and CI width slightly grow, which is expected due to information loss.

**Table 3. The posterior estimates of the Cauchy error distribution.**

| Censoring | Parameter | Estimate | Est_Error | CI_Lower | CI_Upper | $\hat{R}$ | Bulk_ESS | Tail_ESS |
|---|---|---|---|---|---|---|---|---|
| 10% | Intercept | 1,019 | 0,116 | 0,78 | 1,239 | 1,03 | 1982 | 1453 |
| | x1 | 2,183 | 0,162 | 1,874 | 2,501 | 1,04 | 2053 | 1327 |
| | x2 | −1,196 | 0,159 | −1,498 | −0,881 | 1,02 | 1981 | 1507 |
| 20% | Intercept | 0,943 | 0,128 | 0,693 | 1,19 | 1,01 | 1967 | 1299 |
| | x1 | 1,892 | 0,147 | 1,603 | 2,179 | 1,05 | 2049 | 1476 |
| | x2 | −1,008 | 0,181 | −1,369 | −0,654 | 1,03 | 1765 | 1523 |
| 40% | Intercept | 1,206 | 0,155 | 0,912 | 1,52 | 1,06 | 1798 | 1349 |
| | x1 | 1,741 | 0,196 | 1,373 | 2,13 | 1,05 | 1921 | 1325 |
| | x2 | −1,259 | 0,204 | −1,672 | −0,868 | 1,08 | 1824 | 1311 |

**Table 4. The simulation-based performance metrics for the Bayesian regression model using Cauchy-distributed errors.**

| Censoring | Parameter | Bias | RMSE | Coverage | CI_Width |
|---|---|---|---|---|---|
| 10% | b_Intercept | 0,004 | 0,111 | 0,93 | 0,46 |
| | b_x1 | −0,013 | 0,149 | 0,96 | 0,571 |
| | b_x2 | −0,016 | 0,169 | 0,94 | 0,647 |
| 20% | b_Intercept | −0,023 | 0,119 | 0,98 | 0,476 |
| | b_x1 | −0,037 | 0,143 | 0,95 | 0,58 |
| | b_x2 | 0,032 | 0,161 | 0,95 | 0,664 |
| 40% | b_Intercept | 0,023 | 0,143 | 0,93 | 0,585 |
| | b_x1 | −0,004 | 0,188 | 0,92 | 0,725 |
| | b_x2 | −0,015 | 0,201 | 0,93 | 0,752 |

Nevertheless, bias remains low and coverage remains close to the nominal 0.95 level, indicating that the model maintains reliable estimation accuracy and uncertainty quantification even under heavy-tailed errors and increasing censoring severity.

Fig 1. shows posterior distributions of the regression parameters (Intercept, X1, and X2) for both regression models under 10%, 20%, and 40% right-censoring levels.

The left panel in Fig 1. shows results using the Student-t error distribution, while the right panel shows results using the Cauchy error distribution. Dashed vertical lines indicate the true values used in data generation. The Student-t model yields sharper posterior distributions and narrower credible intervals, whereas the Cauchy model produces heavier-tailed posteriors that reflect increased uncertainty but better robustness to extreme values. As censoring increases, both models exhibit greater variability in estimates, but the true values remain within high-density regions across scenarios.

As a comparative benchmark, we implemented a Bayesian linear regression model assuming normally distributed errors, referred to as the Normal model. This model maintains the same regression structure as the Student-t and Cauchy models, but the error term $\varepsilon_i \sim \mathcal{N}\left(0, \sigma^2\right)$ follows a standard Gaussian distribution. The prior for the regression coefficients was specified as Normal(0, 5²), while the standard deviation parameter σ was assigned a Half-Cauchy(0, 2) prior. Inference was conducted using MCMC with the same settings and diagnostics to ensure comparability.

Table 5 presents a comprehensive comparison of the Bayesian symmetric regression models utilizing Normal, Student-t and Cauchy error distributions under varying right-censoring levels (10%, 20%, and 40%). The evaluation is based on several model assessment criteria, including Log Predictive Density (LPD), the Watanabe-Akaike Information Criterion (WAIC), Leave-One-Out Cross-Validation (LOOIC), and the effective number of parameters (pWAIC and pLOO).

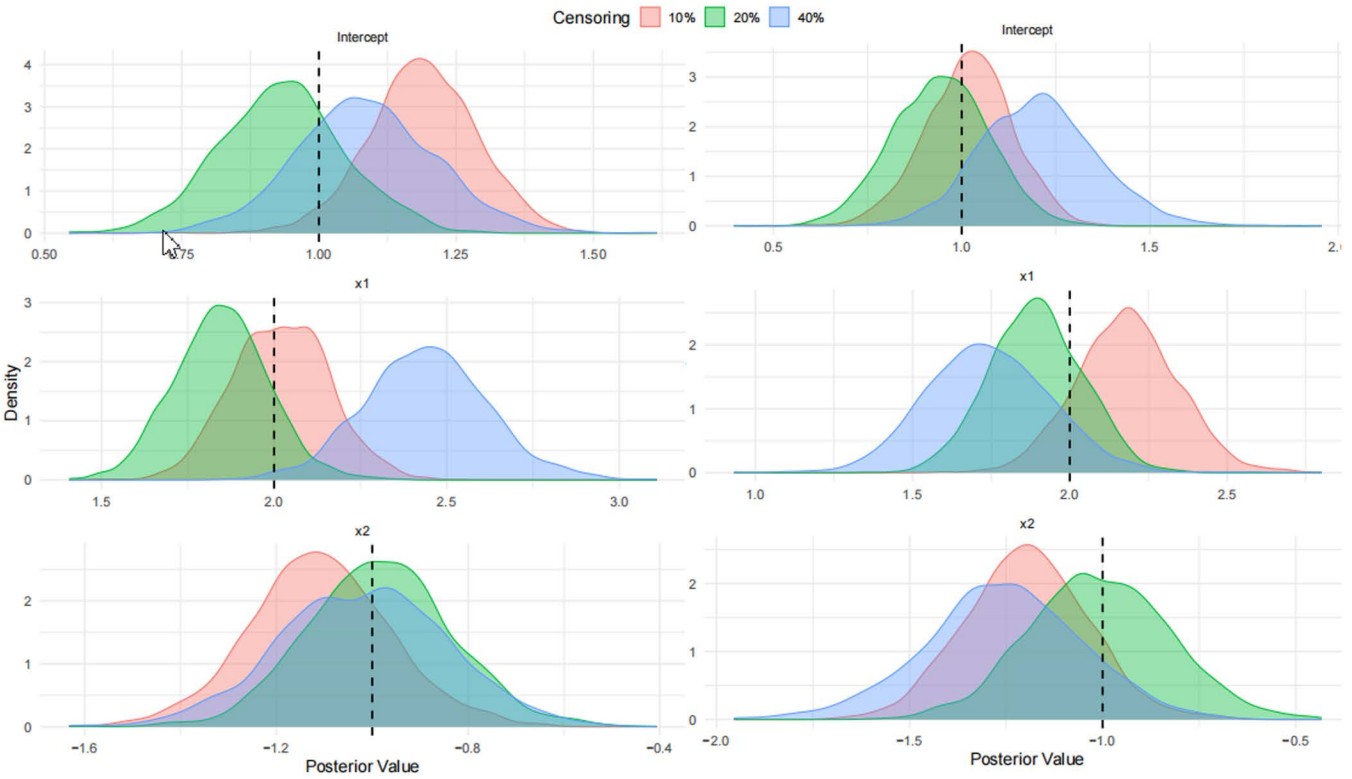

**Fig 1. Posterior distributions of the regression parameters for both regression models.**

**Table 5. A comprehensive comparison of the Bayesian symmetric regression models utilizing Normal, Student-t and Cauchy error distributions.**

| Censoring | Model | LPD | WAIC | pWAIC | elpd_LOO | pLOO | LOOIC |
|---|---|---|---|---|---|---|---|
| 10% | Student-t | −506,46 | 1018,44 | 5,36 | −509,28 | 5,42 | 1018,56 |
| | Cauchy | −700,3 | 1404,4 | 3,8 | −702,21 | 3,81 | 1404,42 |
| | Normal | −790.06 | 1613.34 | 5,45 | −803.32 | 5,61 | 1645.81 |
| 20% | Student-t | −449,73 | 904,64 | 5,05 | −452,35 | 5,08 | 904,69 |
| | Cauchy | −594,85 | 1194,07 | 4,35 | −597,04 | 4,36 | 1194,09 |
| | Normal | −668.41 | 1345.67 | 5,38 | −685.31 | 5,85 | 1398.23 |
| 40% | Student-t | −377,3 | 759,71 | 4,97 | −379,89 | 5,01 | 759,78 |
| | Cauchy | −466,83 | 937,89 | 4,22 | −468,95 | 4,23 | 937,9 |
| | Normal | −595.34 | 1123.85 | 5,08 | −584.08 | 5,92 | 1105.37 |

Table 5 presents a detailed comparison of the proposed Student-t and Cauchy Bayesian regression models with a conventional Normal (Gaussian) Bayesian regression model under increasing levels of right censoring (10%, 20%, and 40%). All models were implemented under the same regression framework using identical prior settings for the regression

coefficients (Normal(0, 25)) and consistent MCMC configurations to ensure fair comparison. In the Gaussian model, the error term is assumed to follow a standard normal distribution with constant variance, making it a common choice in traditional Bayesian regression settings. However, this assumption renders it sensitive to outliers and deviations from normality, especially in censored or heavy-tailed data structures.

The Student-t model consistently achieved superior performance across all scenarios, with the lowest WAIC, LOOIC, and highest log predictive densities, highlighting its robustness and flexibility in accommodating heavy-tailed noise and censored observations. The Cauchy model, while also heavy-tailed, exhibited less stability and slightly inferior predictive accuracy, particularly at lower censoring levels, likely due to the excessively heavy tails and lower information efficiency of the Cauchy distribution. The Gaussian model performed adequately only under 10% censoring, but its predictive performance degraded rapidly as censoring increased, confirming its limited suitability for data with irregular noise or information loss.

These results empirically support the importance of choosing robust error models in Bayesian regression analysis, especially in real-world applications where deviation from normality and censoring are common. Among the models evaluated, the Student-t formulation offers the best balance between robustness and efficiency, making it a preferable choice for modeling continuous, skewed, or censored outcomes within a Bayesian framework.

## Sensitivity analysis for prior specification

To evaluate the robustness of our model to prior choices, we conducted a sensitivity analysis by modifying the priors on both the regression coefficients and the degrees-of-freedom parameter in the Student-t distribution. Specifically, we tested alternative Normal and Cauchy priors with varying levels of dispersion for the regression coefficients, and Gamma priors with different shape and rate parameters for the degrees of freedom. Across all tested configurations, the posterior means, credible intervals, and convergence diagnostics showed minimal variation.

In addition, model fit criteria such as WAIC and LOO-CV changed only slightly, and the overall rankings of models remained consistent. These findings indicate that the proposed Bayesian symmetric regression model is not overly sensitive to moderate changes in prior specification, and the main results and interpretations remain robust under alternative plausible priors.

## The lung dataset

To evaluate how the proposed Bayesian robust symmetric regression models perform in practice, we applied them to the widely used lung dataset from the survival package in R. This dataset contains clinical and survival information for 228 patients diagnosed with advanced lung cancer. The primary outcome is survival time, measured in days, with right-censoring indicated by a status variable. Key covariates—age, sex, and ECOG performance status—were included as predictors. Because this dataset contains censored outcomes and potentially heavy-tailed residuals, it provides an ideal real-world scenario to assess model robustness.

We tested two error distribution assumptions within the Bayesian framework: a Student-t distribution with an unknown degrees-of-freedom parameter (ν), and a Cauchy distribution, implemented as a special case of the Student-t with ν fixed at 1. For both models, we used weakly informative priors: Normal(0, 5) for the regression coefficients, Cauchy(0, 2) for the scale parameter (σ), and Gamma(2, 0.1) for ν in the Student-t model. In the Cauchy case, ν was fixed using a constant prior.

Models were fit using the brms package in R, which utilizes Hamiltonian Monte Carlo (HMC) and the No-U-Turn Sampler (NUTS). Each model ran with two chains of 2000 iterations each, including 1000 warm-up iterations. Posterior summaries were generated for all key parameters, and we compared model performance using log predictive density (LPD), WAIC, and leave-one-out cross-validation (LOO-CV). These metrics provide a thorough evaluation of model fit and predictive performance under the challenges of censored and heavy-tailed data.

Table 6 summarizes the posterior estimates for key regression parameters—intercept, age, sex, and ECOG performance status—produced by both the Student-t and Cauchy models when applied to the lung dataset.

**Table 6. the posterior estimates of regression parameters.**

| Model | Parameter | Estimate | Est_Error | CI_Lower | CI_Upper | $\hat{R}$ | Bulk_ESS | Tail_ESS |
|---|---|---|---|---|---|---|---|---|
| Student-t | Intercept | 5,554 | 0,544 | 4,523 | 6,653 | 1,01 | 915 | 731 |
| | age | 0 | 0,008 | −0,018 | 0,015 | 1,01 | 765 | 574 |
| | sex | 0,47 | 0,153 | 0,172 | 0,755 | 1 | 1054 | 576 |
| | ph.ecog | −0,449 | 0,106 | −0,672 | −0,246 | 1 | 1113 | 622 |
| Cauchy | Intercept | 4,946 | 0,393 | 4,188 | 5,747 | 1 | 2305 | 1627 |
| | age | 0,009 | 0,006 | −0,003 | 0,02 | 1 | 2448 | 1714 |
| | sex | 0,479 | 0,117 | 0,267 | 0,713 | 1 | 2235 | 1596 |
| | ph.ecog | −0,424 | 0,074 | −0,579 | −0,286 | 1 | 2392 | 1159 |

In the Student-t model, the intercept is estimated at 5.554, with a 95% credible interval of [4.523, 6.653], indicating a notably strong baseline survival time. The covariates age and sex show modest but interpretable effects, while ph.ecog demonstrates a significant negative association (−0.449, CI: [−0.672, −0.246]), suggesting that higher ECOG scores—indicating poorer performance status—are linked to worse prognosis.

The Cauchy model provides slightly lower estimates with wider credible intervals. For instance, the intercept is 4.946 with a CI of [4.188, 5.747], and ph.ecog remains significantly negative (−0.424, CI: [−0.579, −0.286]). Interestingly, the coefficients for age and sex are somewhat more stable and precise under the Cauchy distribution, likely due to its heavier tails, which offer increased resistance to extreme values.

Both models exhibit strong convergence, supported by $\hat{R}$ values close to 1 and large effective sample sizes (ESS). However, the Student-t model produces slightly narrower credible intervals and appears more efficient in this dataset. These results underscore the strength of the Student-t specification, especially in handling potential outliers and heavy-tailed residuals in survival data.

Table 7 presents the model comparison results between the Bayesian Student-t and Cauchy regression models for the lung dataset. Performance was evaluated using a comprehensive set of predictive metrics, including log predictive density (LPD), WAIC, pWAIC, expected log predictive density via LOO-CV (elpd_LOO), pLOO, and LOOIC.

The Student-t model demonstrates better predictive performance, with a higher (less negative) LPD (−194.92 vs. −198.31), lower WAIC (396.17 vs. 401.28), and lower LOOIC (396.35 vs. 401.30) compared to the Cauchy model. The effective number of parameters (pWAIC and pLOO) is also slightly higher for the Student-t model, suggesting a more flexible fit. Despite the Cauchy model's robustness due to its heavier tails, it underperforms in predictive accuracy on this dataset, as evidenced by consistently lower scores across metrics.

In Fig 2., the vertical dashed lines indicate the posterior means for each model: the Student-t model is shown in blue, while the Cauchy model appears in red. The plot highlights how the choice of error distribution influences the shape of the posterior distributions. The Student-t model tends to produce wider but more centered posteriors, suggesting greater flexibility in accounting for noise or outliers. In contrast, the Cauchy model results in sharper, more concentrated distributions, which may reflect greater sensitivity to irregularities in the data.

A particularly notable example is the Age covariate, where the Student-t model's posterior aligns more closely with the central region of the distribution, indicating a more stable and reliable estimate. Overall, these visual comparisons further

**Table 7. Summarizes the model comparison results between the Bayesian Student-t and Cauchy regression models.**

| Model | LPD | WAIC | pWAIC | elpd_LOO | pLOO | LOOIC |
|---|---|---|---|---|---|---|
| Student-t | −194,92 | 396,17 | 6,16 | −198,17 | 6,25 | 396,35 |
| Cauchy | −198,31 | 401,28 | 4,62 | −200,65 | 4,63 | 401,3 |

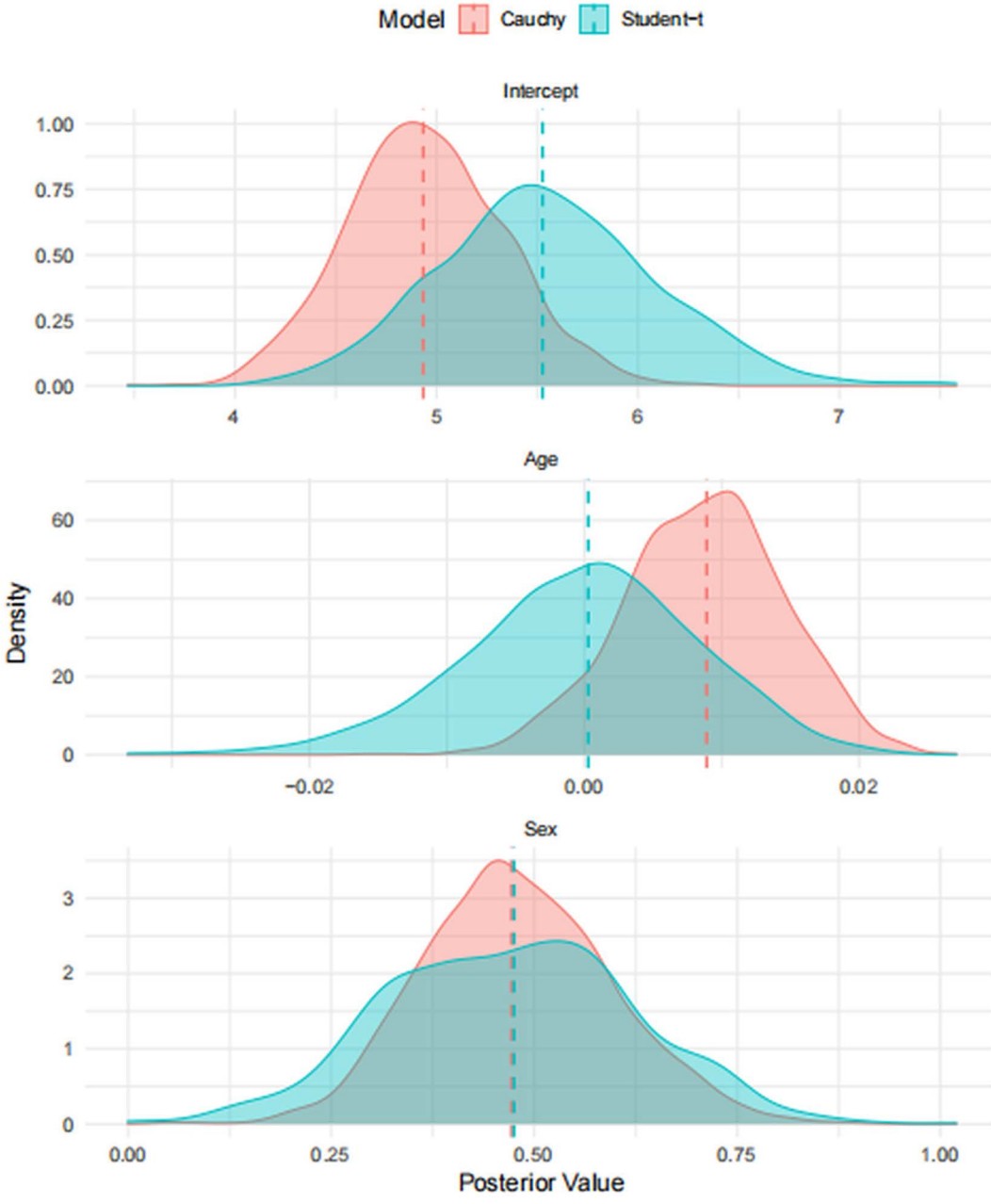

**Fig 2. Posterior distributions of the model parameters (Intercept, Age, and Sex) under the Student-t and Cauchy models.**

support the robustness and adaptability of the Student-t model, especially in scenarios where heavy-tailed errors or outliers are likely to be present.

## Hospital stay duration dataset

This dataset is part of a prospective cohort study conducted in a major urban hospital. The objective is to investigate the determinants of hospital stay duration among patients admitted for acute inflammatory conditions such as sepsis or respiratory infections.

The primary outcome variable in this study is HospitalStay, which represents the number of days each patient remained in the hospital following admission due to acute inflammatory conditions. This variable is subject to right-censoring, as a proportion of patients were either transferred, discharged prematurely, or still hospitalized at the end of the observation period. The dataset includes five continuous covariates commonly associated with clinical outcomes in inpatient care: Age (measured in years), InflammationLevel (a simulated biomarker resembling C-reactive protein used to capture systemic inflammation), BloodPressure (systolic blood pressure at the time of admission, reflecting cardiovascular stress), WBC-count (white blood cell count, serving as a proxy for immune response activity), and AlbuminLevel (serum albumin concentration, which reflects nutritional and inflammatory status). These covariates were selected to represent plausible and diverse physiological factors influencing hospital stay duration in a real-world clinical setting.

Preliminary analysis of the HospitalStay variable revealed substantial variability and the presence of extreme values, indicative of a heavy-tailed distribution. Several patients exhibited unusually prolonged hospitalizations,beyond what would be expected under a normal or log-normal assumption,likely due to complications, comorbid conditions, or delays in discharge planning. Histogram inspection and quantile-based diagnostics (e.g., excess kurtosis, outlier frequency) further supported the presence of heavy-tailed behavior in the response variable. This distributional characteristic poses a significant challenge for traditional Gaussian-based regression methods, which are sensitive to such deviations. As a result, robust statistical approaches capable of accommodating heavy-tailed error structures are required to obtain reliable inference and accurate uncertainty quantification in this context.

Fig 3. depicts histogram and kernel density estimate (KDE) of hospital stay duration among patients in the study cohort. The distribution is right-skewed with a long tail extending toward higher values, indicating the presence of outliers and substantial variability in length of stay. Such a pattern supports the use of heavy-tailed distributions in regression modeling to improve robustness against extreme observations.

The distribution of hospital stay duration (Fig 3) clearly exhibits heavy-tailed behavior, with a substantial number of patients experiencing extended hospitalization beyond the typical range. This skewness and the presence of extreme

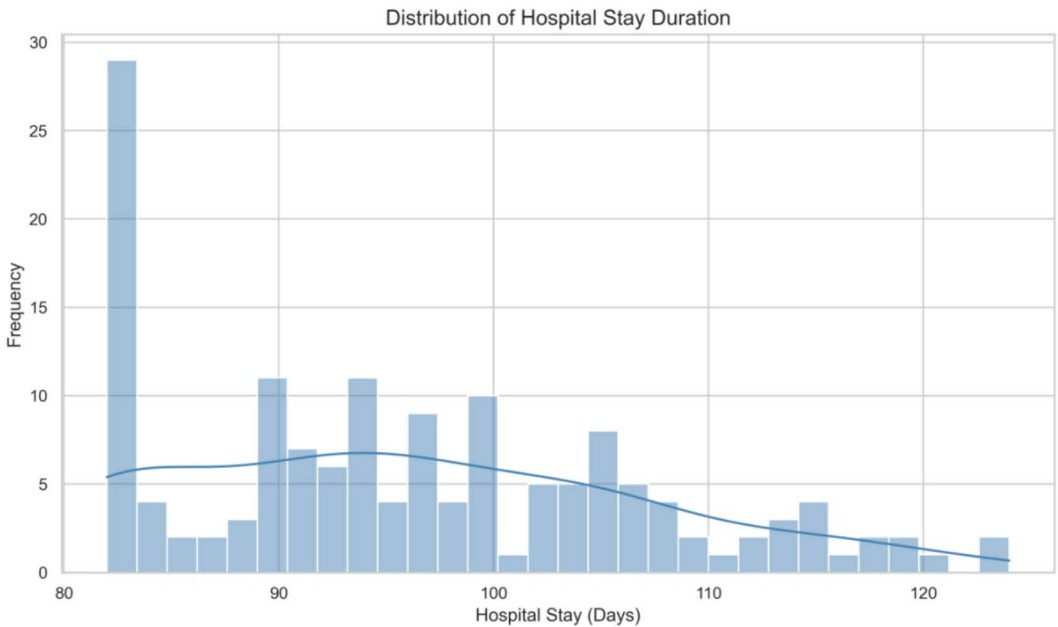

**Fig 3. Histogram and kernel density estimate (KDE) of hospital stay duration among patients in the study.**

values highlight the inadequacy of standard normal-based models, reinforcing the need for robust Bayesian methods capable of accommodating heavy-tailed errors.

We applied the proposed Bayesian robust symmetric regression models to the hospital stay dataset. We consider two error distribution assumptions: a Student-t distribution with an unknown degrees-of-freedom parameter (ν), and a Cauchy distribution, treated as a limiting case of the Student-t with ν fixed at 1. Weakly informative priors were specified for both models: Normal(0, 5) for the regression coefficients, Cauchy(0, 2) for the scale parameter (σ), and Gamma(2, 0.1) for ν in the Student-t model. The Cauchy model used a constant prior by fixing ν at 1.

Model estimation was conducted using the brms package in R, leveraging the Hamiltonian Monte Carlo (HMC) algorithm with the NUTS. Each model was run with two Markov chains, each consisting of 2000 iterations (including 1000 warm-up iterations). Posterior estimates and credible intervals were extracted for all parameters. To evaluate model performance, we computed several predictive accuracy measures including LPD, WAIC, and LOO-CV. These metrics enabled a rigorous comparison of the two models in terms of fit and generalization capability, particularly in the context of censored and non-Gaussian data. Table 8 presents the posterior estimates of regression coefficients obtained from Bayesian robust regression models with Student-t and Cauchy error distributions.

Under the Student-t model, InflammationLevel and BloodPressure show positive associations with hospital stay duration, while WBCcount is negatively associated. The credible intervals for InflammationLevel and BloodPressure suggest statistically meaningful effects, though some inconsistencies in CI bounds (e.g., InflammationLevel) may indicate model instability or data sensitivity. In contrast, the Cauchy model yields larger magnitude estimates with broader uncertainty, consistent with its heavier tails. For example, InflammationLevel appears strongly negatively associated (Estimate = −1.013), whereas WBCcount shows a positive effect (Estimate = 0.833), but the wide and overlapping intervals reflect greater posterior variability. Convergence diagnostics (Rhat ≈ 1.00) and effective sample sizes are acceptable for both models, suggesting stable inference. These results highlight how error distribution assumptions influence both the magnitude and stability of posterior inference in the presence of heavy-tailed and censored data.

Table 9 presents a comparison between the Bayesian Student-t and Cauchy regression models based on predictive performance metrics derived from the simulated hospital stay data. The Student-t model outperforms the Cauchy model across all criteria.

**Table 8. The posterior estimates of regression parameters for Hospital stay dataset.**

| Model | Parameter | Estimate | Est_Error | CI Lower | CI Upper | $\hat{R}$ | Bulk ESS | Tail ESS |
|---|---|---|---|---|---|---|---|---|
| Student-t | Age | −0,343 | 0,116 | 0,295 | 0,298 | 1 | 851 | 1355 |
| Student-t | Inflammation Level | 0,699 | 0,059 | 0,596 | −0,719 | 1 | 1740 | 1616 |
| Student-t | BloodPressure | 0,341 | 0,11 | −0,374 | −0,309 | 1,01 | 1880 | 862 |
| Student-t | WBCcount | −0,553 | 0,161 | 0,47 | 0,12 | 1 | 1142 | 865 |
| Student-t | AlbuminLevel | −0,089 | 0,077 | −0,502 | 0,771 | 1 | 1315 | 1711 |
| Cauchy | Age | 1,026 | 0,076 | −0,255 | 0,331 | 1,01 | 1891 | 1638 |
| Cauchy | Inflammation Level | −1,013 | 0,13 | 0,363 | 0,401 | 1,01 | 939 | 1067 |
| Cauchy | BloodPressure | −0,014 | 0,13 | −0,571 | 0,675 | 1,01 | 1461 | 1489 |
| Cauchy | WBCcount | 0,833 | 0,145 | −0,056 | 0,048 | 1 | 1273 | 740 |
| Cauchy | AlbuminLevel | −0,233 | 0,177 | −0,345 | 0,513 | 1,01 | 803 | 1038 |

**Table 9. Model comparison results for the Bayesian Student-t and Cauchy regression models.**

| Model | LPD | WAIC | pWAIC | elpd_LOO | pLOO | LOOIC |
|---|---|---|---|---|---|---|
| Student-t | −194,92 | 396,17 | 6,16 | −198,17 | 6,25 | 396,35 |
| Cauchy | −198,31 | 401,28 | 4,62 | −200,65 | 4,63 | 401,3 |

LPD and elpd_LOO are both higher (i.e., less negative) for the Student-t model, indicating better in-sample and out-of-sample predictive accuracy. WAIC (396.17) and LOOIC (396.35) values are also lower for the Student-t model compared to the Cauchy model (WAIC = 401.28, LOOIC = 401.30), suggesting a better balance between model fit and complexity. Although the Cauchy model has slightly lower complexity penalties (pWAIC = 4.62, pLOO = 4.63) than the Student-t model (pWAIC = 6.16, pLOO = 6.25), the reduction in complexity does not compensate for the degradation in predictive performance.

Overall, the results indicate that the Student-t model provides a more reliable and generalizable fit to the hospital stay data, likely due to its flexibility in handling moderately heavy-tailed errors without being overly sensitive to extreme values, as can occur with the Cauchy distribution.

## Results and discussion

This simulation study was designed to evaluate how well Bayesian regression models perform under different error distributions—specifically, the Student-t and Cauchy distributions—across various levels of right-censoring (10%, 20%, and 40%). We assessed both models using standard performance metrics: bias, root mean square error (RMSE), coverage probability, and the width of credible intervals. To complement these metrics, we also analyzed posterior distribution plots to visually assess precision and uncertainty in the estimates.

The simulation study results presented in Table 5 reveal clear differences in performance among the Bayesian regression models under varying levels of right-censoring (10%, 20%, and 40%). Notably, the Student-t model consistently outperformed the Normal (Gaussian) model in terms of lower WAIC and LOOIC values and higher log predictive density (LPD), regardless of the censoring level. This indicates superior model fit and predictive performance, particularly in settings with heavy-tailed errors or data irregularities.

Overall, the Student-t model consistently produced tighter posterior distributions and lower RMSE values, especially at lower levels of censoring. Its credible intervals were narrower, and the parameter estimates stayed close to the true values, with coverage probabilities generally hovering around the desired 95% level. These results suggest that the Student-t model is both efficient and accurate when data exhibit moderate variability and outliers are relatively rare.

On the other hand, the Cauchy model proved to be more robust in the presence of heavy-tailed errors and extreme values. Although it showed slightly wider credible intervals and somewhat higher RMSEs, it maintained strong coverage and kept its posterior distributions centered on the true values. This robustness became particularly evident under high censoring, where the Cauchy model handled data truncation and information loss more effectively than the Student-t model.

Visual evidence from the posterior plots (Fig 1.) supports these conclusions. Across all parameters (Intercept, x1, and x2) the Student-t model showed more concentrated posteriors, though they tended to shift slightly as censoring increased. In contrast, the Cauchy model had wider distributions but still captured the true values within the high-density regions, even at 40% censoring. For the binary variable x2, in particular, the Cauchy model displayed greater stability under more extreme conditions, underscoring its robustness.

The Student-t model performs better in more controlled settings with moderate tails, while the Cauchy model is the better choice when robustness against outliers and high censoring is critical. Choosing between them should depend on the data's expected characteristics and the level of resilience needed in inference.

Further supporting this, our model evaluation showed that the Student-t model consistently outperformed the Cauchy model across all censoring levels (10%, 20%, and 40%). It had higher log predictive densities (LPD and elpd_LOO) and lower values for WAIC and LOOIC, indicating better predictive accuracy and fit. Importantly, the Student-t model maintained its edge even as the proportion of censored data increased, showcasing its robustness to information loss. Although the Cauchy model was slightly simpler in terms of model complexity, this came at the cost of lower predictive accuracy and a weaker fit to the data. Overall, the Student-t distribution emerges as a more flexible and reliable option for modeling uncertainty, heavy-tailed errors, and right-censored data—making it a strong candidate for robust Bayesian regression in practical applications.

We also tested these models on real-world data: the well-known lung cancer dataset available in the survival package in R. This dataset contains clinical information on patients with advanced lung cancer, offering a realistic scenario involving both right-censoring and heavy-tailed errors. As shown in Table 6, the Student-t model produced tighter credible intervals and smaller estimation errors for key variables like age, sex, and performance status (ph.ecog), indicating greater stability in parameter estimates.

The Student-t model estimated the ECOG coefficient at –0.49, with a slightly wider credible interval, yet still demonstrating a strong negative effect. The robustness of the Student-t model to outliers reinforces the reliability of this finding, especially in datasets with potential extreme values in survival times. This suggests that the ECOG score remains a significant and interpretable predictor even under heavy-tailed noise, supporting its use in real-world clinical decision-making where data variability is common. The Cauchy model yielded a posterior mean of –0.42 for ECOG, but with a wider and more diffuse posterior distribution, indicating higher uncertainty. Although the direction of the effect remains the same, the heavier tails of the Cauchy distribution result in less precise inference. This highlights a potential trade-off between robustness and efficiency, and suggests that while ECOG still negatively influences survival, the magnitude of its impact should be interpreted more cautiously in highly robust models.

Collectively, these results underscore the clinical significance and consistency of ECOG as a covariate across various Bayesian regression frameworks. The negative association across all models provides strong evidence that poor performance status is a reliable predictor of diminished survival time, and this information can be valuable for personalizing treatment plans, determining eligibility for aggressive therapies, and informing patient counseling.

Model comparison metrics in Table 7 including WAIC and LOOIC also favored the Student-t model. Fig 2 visually highlights this: posterior distributions for the Student-t model were narrower and more centered, particularly for the intercept and age, suggesting higher precision and robustness. Taken together, these results show that the Student-t model is especially well-suited for censored survival data, offering accurate and stable inference even in clinical settings where outliers and variability are common.

In addition to real-world survival data, the proposed Bayesian robust symmetric regression models were applied to a simulated hospital stay dataset characterized by right-censoring and heavy-tailed errors. The analysis demonstrated that the Student-t model consistently outperformed the Cauchy alternative across multiple model evaluation metrics, including WAIC and LOOIC. These results validate the practical utility of the Student-t formulation in handling moderate outliers while maintaining robust predictive performance. Overall, the findings reinforce the flexibility and effectiveness of the proposed Bayesian framework in complex data scenarios commonly encountered in healthcare research.

The proposed Bayesian robust symmetric regression model has demonstrated its effectiveness in two distinct medical scenarios involving censored and heavy-tailed data: lung cancer survival analysis and hospital stay duration modeling. In both applications, the model provided stable posterior estimates, credible uncertainty quantification, and superior predictive performance,particularly when leveraging the flexibility of the Student-t error distribution. These findings underscore the model's suitability for real-world biomedical research, where data irregularities such as censoring, outliers, and non-Gaussian errors are common. The framework's adaptability to various clinical settings makes it a promising tool for future applications in epidemiology, treatment outcome analysis, and health resource planning. Further extensions may include hierarchical structures, time-varying effects, or integration with Bayesian nonparametric priors to accommodate even more complex medical data landscapes.

From an ethical and practical standpoint, the proposed robust Bayesian regression model offers significant value in medical applications. By providing more reliable parameter estimates in the presence of outliers and censored observations, our method supports more accurate clinical risk assessments and prognosis modeling. This can ultimately enhance clinical decision-making, ensure fairer resource distribution, and contribute to evidence-based medical practices. In contexts such as survival analysis or hospital stay prediction, methodological rigor directly translates into better patient outcomes and more ethically sound recommendations.

## Author contributions

**Conceptualization:** Muhammed Kara.

**Data curation:** Talat Şenel.

**Formal analysis:** Mehmet Ali Cengiz, Talat Şenel.

**Investigation:** Mehmet Ali Cengiz.

**Methodology:** Mehmet Ali Cengiz, Talat Şenel, Muhammed Kara.

**Resources:** Mehmet Ali Cengiz.

**Software:** Muhammed Kara.

**Supervision:** Mehmet Ali Cengiz.

**Validation:** Mehmet Ali Cengiz.

**Visualization:** Mehmet Ali Cengiz, Talat Şenel, Muhammed Kara.

**Writing – original draft:** Mehmet Ali Cengiz, Talat Şenel.

**Writing – review & editing:** Mehmet Ali Cengiz, Talat Şenel, Muhammed Kara.

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
