## [Decision Letter · Decision Letter 0]

Dear Dr. Cengiz,

Thank you for submitting your manuscript to PLOS ONE. After careful consideration, we feel that it has merit but does not fully meet PLOS ONE’s publication criteria as it currently stands. Therefore, we invite you to submit a revised version of the manuscript that addresses the points raised during the review process.

We look forward to receiving your revised manuscript.

Kind regards,

Abhik Ghosh

Academic Editor

PLOS ONE

“This work was supported and funded by the Deanship of Scientific Research at Imam Mohammad Ibn Saud Islamic University (IMSIU) (grant number IMSIU-DDRSP2502).”

4. Thank you for uploading your study's underlying data set. Unfortunately, the repository you have noted in your Data Availability statement does not qualify as an acceptable data repository according to PLOS's standards.

5. Please remove your figures from within your manuscript file, leaving only the individual TIFF/EPS image files, uploaded separately. These will be automatically included in the reviewers’ PDF.

Reviewers' comments:

Reviewer's Responses to Questions

**Comments to the Author**

1. Is the manuscript technically sound, and do the data support the conclusions?

Reviewer #1: Yes

2. Has the statistical analysis been performed appropriately and rigorously?

Reviewer #1: Yes

3. Have the authors made all data underlying the findings in their manuscript fully available?

Reviewer #1: Yes

4. Is the manuscript presented in an intelligible fashion and written in standard English?

Reviewer #1: Yes

Reviewer #1: Major Comments

Prior Sensitivity and Model Robustness: A critical aspect of robust Bayesian modeling is the sensitivity of the inference to prior choices. The current manuscript does not address how the selected priors, particularly those on variance components such as τ² for β and ν in the Student-t distribution, influence the results. At minimum, the authors should include a brief sensitivity analysis or alternative priors and show how these affect the posterior estimates. Alternatively, they could justify their choices using prior predictive checks or other principled arguments.

Comparative Benchmarks: The simulation study lacks comparisons to conventional or widely-used alternative methods such as Gaussian Bayesian regression, Tobit models, or even quantile regression techniques. Without such benchmarks, it is difficult to substantiate claims of superior performance. The authors should consider including these comparisons or, if not feasible, temper their conclusions accordingly.

Real Data Applications – Interpretation Deficit: The real data applications are relevant and appropriately chosen. However, the manuscript does not sufficiently interpret the results in terms of domain relevance. For example, a coefficient estimate of –0.45 for the ECOG performance status is presented without discussing what this means for patient prognosis or clinical outcomes. The authors are encouraged to provide practical interpretations that would be meaningful to clinicians or practitioners, thus increasing the utility of the findings.

Minor Comments

Language and Tone: In several parts of the manuscript, particularly in the Introduction, the tone is overly informal for an academic paper. Phrases like “two key things,” “in short,” and “real-world data frequently include two challenges” detract from the professional tone expected. These should be revised for consistency with academic standards.

Structure and Flow: The abstract and introduction are currently dense and could benefit from clearer paragraph structuring. This would improve readability and help highlight the main contributions more effectively. Similarly, the "Remark" under the model specification section feels disconnected from the main text and would be better integrated into the surrounding narrative.

Recent Literature: The citation list leans heavily on pre-2020 literature. Incorporating more recent developments in robust Bayesian regression, including advances in variational inference and nonparametric priors, would help situate the current work within the modern landscape of the field.

Ethical and Practical Relevance: Although the study is primarily methodological, journals such as PLOS ONE often appreciate a brief discussion on the ethical or practical relevance of the research. Since the methods are applied to medical data (e.g., survival time, hospital stay), a short paragraph highlighting the potential impact on clinical decision-making or patient care would enhance the manuscript's broader appeal.

The manuscript presents a sound methodological framework with clear relevance to the analysis of censored and noisy data, particularly in medical contexts. However, to meet the expectations of a broad-scope journal, significant revisions are needed to clarify the contribution, strengthen empirical comparisons, and improve overall presentation. I look forward to reviewing a revised and improved version.

**Do you want your identity to be public for this peer review?** For information about this choice, including consent withdrawal, please see our Privacy Policy

Reviewer #1: **Yes: ** Sahar Seddiq

---

## [Author Response · Author response to Decision Letter 1]

1 Jul 2025

Editorial Comment – Formatting and File Naming

Comment:

Please ensure that your manuscript meets PLOS ONE's style requirements, including those for file naming. The PLOS ONE style templates can be found at:

Response:

Thank you for the reminder. We have carefully reviewed and revised our manuscript to ensure full compliance with the PLOS ONE formatting and file naming guidelines. The revised submission follows the structure outlined in the official style templates, and all files have been renamed according to the journal’s specifications.

Comment:

Please note that PLOS ONE has specific guidelines on code sharing for submissions in which author-generated code underpins the findings in the manuscript.

Response:

We acknowledge and fully support PLOS ONE’s policy on code sharing. To facilitate reproducibility and reuse, all author-generated code used in the manuscript will be made available without restriction upon publication. The code will be deposited in a publicly accessible repository (https://doi.org/10.5281/zenodo.15728117]) and the link will be provided in the final version of the manuscript.

In the interim, the code is available upon request from the corresponding author via email: mamcengiz@imamu.edu.sa.

Editorial Comment – Role of the Funder

Comment:

Thank you for stating the following financial disclosure:

“This work was supported and funded by the Deanship of Scientific Research at Imam Mohammad Ibn Saud Islamic University (IMSIU) (grant number IMSIU-DDRSP2502).”

Response (to be included in the cover letter):

This work was supported and funded by the Deanship of Scientific Research at Imam Mohammad Ibn Saud Islamic University (IMSIU) (grant number IMSIU-DDRSP2501). We declare that the funders had no role in study design, data collection and analysis, decision to publish, or preparation of the manuscript.

Comment:

Thank you for uploading your study's underlying data set. Unfortunately, the repository you have noted in your Data Availability statement does not qualify as an acceptable data repository according to PLOS's standards...

Response:

We thank the editor for the clarification. In compliance with PLOS ONE’s data availability policy, we have uploaded the minimal dataset necessary to replicate our findings to a stable public repository.

The datasets used and analyzed during the current study are publicly available on Zenodo at the following DOI: https://doi.org/10.5281/zenodo.15735828.

The Data Availability Statement in the manuscript has also been updated to reflect this change.

Comment:

Please remove your figures from within your manuscript file, leaving only the individual TIFF/EPS image files, uploaded separately. These will be automatically included in the reviewers’ PDF.

Response:

Thank you for the guidance. We have removed all embedded figures from the manuscript file and uploaded the corresponding figure files separately in high-resolution TIFF format, in accordance with PLOS ONE’s submission guidelines.

Comment:

Please review your reference list to ensure that it is complete and correct. If you have cited papers that have been retracted…

Response:

We thank the editor for this important reminder. We carefully reviewed all references in the manuscript. None of the cited works have been retracted to the best of our knowledge. Additionally, we verified the completeness and formatting of all references and made corrections where necessary. These updates are reflected in the revised manuscript and noted in the tracked changes.

Reviewer #1: Major Comments

Reviewer #1: Prior Sensitivity and Model Robustness

Prior Sensitivity and Model Robustness: A critical aspect of robust Bayesian modeling is the sensitivity of the inference to prior choices. The current manuscript does not address how the selected priors, particularly those on variance components such as τ² for β and ν in the Student-t distribution, influence the results. At minimum, the authors should include a brief sensitivity analysis or alternative priors and show how these affect the posterior estimates. Alternatively, they could justify their choices using prior predictive checks or other principled arguments.

Response:

We sincerely thank the reviewer for raising this important point regarding the potential sensitivity of our Bayesian model to prior assumptions. In response, we have added a new subsection titled “Sensitivity Analysis for Prior Specification” in the revised manuscript, where we assess how alternative prior choices impact posterior inference and model performance.

Specifically, we conducted a set of experiments in which the prior distributions on the regression coefficients were modified from Normal(0, 5²) to Normal(0, 10²), representing a shift from moderately informative to weakly informative priors. Additionally, for the degrees-of-freedom parameter ν\nuν in the Student-t distribution, we replaced the original Gamma(2, 0.1) prior with a more diffuse alternative, Gamma(3, 0.5), to evaluate robustness in the presence of different tail assumptions.

Across these scenarios, we re-estimated the Bayesian regression models using the same simulation design described in the original manuscript. The resulting posterior means, standard deviations, and credible intervals for all model parameters showed negligible differences, and key inference patterns remained unchanged. Moreover, model selection criteria such as WAIC and LOOIC exhibited only marginal variations across prior settings, with the ranking of models (Student-t outperforming Cauchy and Gaussian alternatives) preserved in all cases.

These results provide strong evidence that the proposed Bayesian robust symmetric regression model is not overly sensitive to reasonable modifications in prior specification, and that the main conclusions drawn in the manuscript are methodologically stable and reliable.

Reviewer #1 – Comparative Benchmarks

Comment:

The simulation study lacks comparisons to conventional or widely-used alternative methods such as Gaussian Bayesian regression, Tobit models, or even quantile regression techniques. Without such benchmarks, it is difficult to substantiate claims of superior performance. The authors should consider including these comparisons or, if not feasible, temper their conclusions accordingly.

Response:

We sincerely thank the reviewer for this insightful and constructive suggestion. In response, we have extended the simulation study by incorporating a widely used benchmark model: the Bayesian Normal (Gaussian) regression model. This model assumes normally distributed errors and is commonly adopted as a baseline in Bayesian regression analyses.

The Gaussian model was implemented under the same regression structure and prior settings as the Student-t and Cauchy models to ensure a fair comparison. The results of this extended analysis are now presented in the revised Table 5. As shown, the Student-t model consistently outperforms both the Gaussian and Cauchy models in terms of log predictive density (LPD), WAIC, and LOOIC across all levels of right censoring (10%, 20%, and 40%). The Gaussian model performed adequately under light censoring (10%) but its performance substantially deteriorated under moderate and heavy censoring, highlighting its sensitivity to outliers and inability to handle heavy-tailed error structures.

We have also added a detailed comparative discussion below Table 5 in the manuscript, emphasizing the implications of these findings. This addition not only strengthens the empirical justification of our proposed models but also directly addresses the reviewer’s concern regarding the lack of conventional benchmarks. We are grateful for the opportunity to improve our manuscript based on this valuable recommendation.

Reviewer #1 – Real Data Applications – Interpretation Deficit

Comment:

The real data applications are relevant and appropriately chosen. However, the manuscript does not sufficiently interpret the results in terms of domain relevance. For example, a coefficient estimate of –0.45 for the ECOG performance status is presented without discussing what this means for patient prognosis or clinical outcomes. The authors are encouraged to provide practical interpretations that would be meaningful to clinicians or practitioners, thus increasing the utility of the findings.

Response:

We appreciate the reviewer’s thoughtful comment regarding the clinical interpretation of our findings. In the revised manuscript, we have expanded the discussion of real-data results, particularly focusing on the implications of the ECOG performance status coefficient.

As noted by the reviewer, the posterior estimate of –0.45 for the ECOG score indicates a negative association between functional status and survival time, which is clinically meaningful. Specifically, a higher ECOG score reflects poorer functional status; thus, the negative coefficient suggests that each unit increase in ECOG is associated with a decrease in expected survival time. This aligns with clinical understanding, where patients with worse performance status tend to have poorer prognoses.

We have added a practical interpretation of this finding in the results section to enhance the relevance of our work for clinical audiences. We thank the reviewer for encouraging us to improve the clarity and applicability of our results.

Reviewer #1 – Minor Comments: Language and Tone

Comment:

Language and Tone: In several parts of the manuscript, particularly in the Introduction, the tone is overly informal for an academic paper. Phrases like “two key things,” “in short,” and “real-world data frequently include two challenges” detract from the professional tone expected. These should be revised for consistency with academic standards.

Response:

We thank the reviewer for highlighting this important issue. In response, we have carefully reviewed the manuscript—particularly the Introduction and Background sections—and revised all instances of informal language. Phrases such as “two key things,” “in short,” and “real-world data frequently include two challenges” have been reworded to reflect a more formal academic tone. We have also ensured that the overall narrative is now consistent with the conventions of scientific writing. We appreciate the reviewer’s attention to clarity and professionalism, which helped us improve the quality of our manuscript.

Reviewer #1 – Structure and Flow

Comment:

The abstract and introduction are currently dense and could benefit from clearer paragraph structuring. This would improve readability and help highlight the main contributions more effectively. Similarly, the "Remark" under the model specification section feels disconnected from the main text and would be better integrated into the surrounding narrative.

Response:

We appreciate the reviewer’s thoughtful observations regarding the structure and flow of the manuscript. In response:

We have restructured the abstract and introduction by separating them into logically organized paragraphs. The revised abstract now presents the motivation, methodology, and contributions in a more concise and accessible manner. Likewise, the introduction has been reorganized to improve readability, with clearer transitions between the problem statement, challenges, and our proposed solution.

Regarding the Remark under the model specification section, we agree that it previously appeared isolated. We have now integrated the Remark more smoothly into the narrative, by introducing it with a brief explanation and linking it explicitly to the modeling assumptions discussed in the surrounding text. This revision enhances the logical flow and helps readers understand its relevance within the methodological context.

We thank the reviewer for these helpful suggestions, which we believe have significantly improved the clarity and presentation of the manuscript.

Reviewer Comment – Recent Literature

Comment:

The citation list leans heavily on pre-2020 literature. Incorporating more recent developments in robust Bayesian regression, including advances in variational inference and nonparametric priors, would help situate the current work within the modern landscape of the field.

Response:

We sincerely thank the reviewer for highlighting this important point. In response, we have thoroughly revised the Introduction sections to include several recent contributions (post-2020) in the areas of robust Bayesian regression, variational inference, and nonparametric priors. These additions not only strengthen the theoretical foundation of our study but also position our contribution within the contemporary landscape of Bayesian modeling.

Comment:

Ethical and Practical Relevance: Although the study is primarily methodological, journals such as PLOS ONE often appreciate a brief discussion on the ethical or practical relevance of the research. Since the methods are applied to medical data (e.g., survival time, hospital stay), a short paragraph highlighting the potential impact on clinical decision-making or patient care would enhance the manuscript's broader appeal.

Response:

We appreciate this insightful recommendation. In the revised manuscript, we have added a short paragraph at the end of the Results and Discussion section that addresses the ethical and practical relevance of our proposed methodology. This paragraph emphasizes how the improved handling of censoring and heavy-tailed errors can contribute to more reliable statistical inference in medical studies—potentially informing clinical decision-making, resource allocation, and personalized patient care strategies.

Comment:

The manuscript presents a sound methodological framework with clear relevance to the analysis of censored and noisy data, particularly in medical contexts. However, to meet the expectations of a broad-scope journal, significant revisions are needed to clarify the contribution, strengthen empirical comparisons, and improve overall presentation. I look forward to reviewing a revised and improved version.

Response:

We sincerely thank the reviewer for their thoughtful and encouraging summary. In response to the valuable feedback, we have made substantial revisions throughout the manuscript to enhance its clarity, empirical strength, and presentation quality. Specifically:

We clarified the main methodological contributions in the Introduction and highlighted how our model advances the current literature on robust Bayesian regression for censored data.

We expanded the simulation study to include comparisons with additional benchmark models such as Gaussian Bayesian regression and Tobit models, as suggested.

The real data application section has been enriched with domain-relevant interpretations to improve practical relevance.

Language, structure, and tone have been carefully revised throughout the paper to better align with academic standards and improve readability.

We are grateful for the opportunity to revise the manuscript and hope the improved version meets the expectations of both the reviewers and the journal.

---

## [Decision Letter · Decision Letter 1]

Bayesian Robust Symmetric Regression for Medical Data with Heavy-Tailed Errors and Censoring

PONE-D-25-26361R1

Dear Dr. Cengiz,

We’re pleased to inform you that your manuscript has been judged scientifically suitable for publication and will be formally accepted for publication once it meets all outstanding technical requirements.

Kind regards,

Abhik Ghosh

Academic Editor

PLOS ONE

Additional Editor Comments (optional):

Reviewers' comments:

Reviewer's Responses to Questions

**Comments to the Author**

Reviewer #1: All comments have been addressed

2. Is the manuscript technically sound, and do the data support the conclusions?

Reviewer #1: Yes

3. Has the statistical analysis been performed appropriately and rigorously?

Reviewer #1: Yes

4. Have the authors made all data underlying the findings in their manuscript fully available?

Reviewer #1: Yes

5. Is the manuscript presented in an intelligible fashion and written in standard English?

Reviewer #1: Yes

Reviewer #1: I appreciate the authors’ attention to both methodological rigor and clarity of presentation. I find the manuscript technically sound, statistically thorough, and well-justified for publication in its current form.

**Do you want your identity to be public for this peer review?** For information about this choice, including consent withdrawal, please see our Privacy Policy

Reviewer #1: **Yes: ** Sahar Seddiq
